# PDEC: A Framework for Improving Knowledge Graph Reasoning Performance through Predicate Decomposition

**Xin Tian [1,\*] and Yuan Meng [2]**

1    School of Computer Science, National University of Defense Technology, Changsha 410073, China
2    Department of Computer Science and Technology, Tsinghua University, Beijing 100084, China;
     yuanmeng@tsinghua.edu.cn
*    Correspondence: tianxin07@nudt.edu.cn

**Abstract:** The judicious configuration of predicates is a crucial but often overlooked aspect in the field of knowledge graphs. While previous research has primarily focused on the precision of triples in assessing knowledge graph quality, the rationality of predicates has been largely ignored. This paper introduces an innovative approach aimed at enhancing knowledge graph reasoning by addressing the issue of predicate polysemy. Predicate polysemy refers to instances where a predicate possesses multiple meanings, introducing ambiguity into the knowledge graph. We present an adaptable optimization framework that effectively addresses predicate polysemy, thereby enhancing reasoning capabilities within knowledge graphs. Our approach serves as a versatile and generalized framework applicable to any reasoning model, offering a scalable and flexible solution to enhance performance across various domains and applications. Through rigorous experimental evaluations, we demonstrate the effectiveness and adaptability of our methodology, showing significant improvements in knowledge graph reasoning accuracy. Our findings underscore that discerning predicate polysemy is a crucial step towards achieving a more dependable and efficient knowledge graph reasoning process. Even in the age of large language models, the optimization and induction of predicates remain relevant in ensuring interpretable reasoning.

**Keywords:** knowledge graph; reasoning; predicates; embedding

## 1. Introduction

Predicates play a pivotal role in knowledge management and interpretable reasoning. Formulating a reasonable and precise set of predicates is a fundamental step in building a knowledge graph (KG). Despite the widespread use of large language models (LLMs), a well-configured predicate set remains crucial for achieving interpretable reasoning.

Traditional KG-based reasoning research typically operates in a transductive setting, where the triples to predict involve only entities and predicates that have occurred in the embedding training triples [1–4]. This is known as transductive knowledge graph completion (*transductive KGC*). Conversely, *inductive KGC* aims to complete triples involving unseen entities without retraining the KG embeddings from scratch [5,6]. However, most inductive KGC studies focus on introducing new entities in the test set, rather than new predicates [7,8]. This assumes that the predicate set configuration is reasonable, which is often not the case.

Predicates in a KG may possess multiple semantics, reflected in the local structure associated with the predicate in the graph [9]. When a predicate exhibits multiple patterns in its local structure, it may have multiple semantics. For example, in Freebase [10], the predicate `Person-Language` represents the relationship between persons and the language they speak. This relationship can be further subcategorized into predicate relationships between persons and their mother tongue (`Person-NativeLanguage`) and between persons and the foreign language they master (`Person-ForeignLanguage`). The former exhibits local

structural features closely associated with predicates like `Person-Nationality`. Predicates with multiple semantics can be termed as *polysemous predicates*.

Additionally, there may be issues of *synonymous predicates*, where multiple distinct predicates share identical semantic features, leading to confusion in distinguishing between them. Conversely, there may be issues of *predicates missing* in KGs, where a particular local structural pattern between entities is not captured by a single predicate (e.g., a KG has the predicate `husbandOf` but lacks the predicate `coupleOf`; the latter could assist KGC tasks related to marital relationships on the KG).

These issues of predicate set configuration can significantly impact the ability of KGs to represent knowledge and the performance of reasoning models. Therefore, it is crucial to address these issues to enhance the quality of KGs and improve the performance of reasoning models.

Recently, some research has been dedicated to implementing *predicate inductive KGC*, which involves introducing predicates that have never existed in the training set into the test set of KGC tasks [11]. However, these efforts either rely on additional ontology information and textual information for assistance in induction or focus solely on feasibility studies, often sacrificing the performance of reasoning models [12]. Furthermore, they have primarily considered the issue of missing predicates and have not delved into issues of polysemy and synonymy of predicates.

In contrast, our work focuses on addressing the problem of polysemous predicates and proposes a simple yet effective method, PDEC (predicates <u>dec</u>omposition), to enhance reasoning model performance without introducing any external information. PDEC optimizes the quality of KGs by decomposing polysemous predicates into more specific ones, thereby improving the accuracy and interpretability of reasoning.

Our approach is based on representation learning, where we obtain entity and predicate embeddings by calculating *edge vectors* for each triplet in the training set. These vectors are then clustered to identify polysemy in predicates and determine an optimal split for the polysemous predicates. This refined KG, achieved through *predicate decomposition*, can be seamlessly integrated into any reasoning model. Moreover, the original predicates can be reconstructed from the split predicates, enabling the model to operate on the original KG. In addition, we explore other challenges related to predicate set configuration, such as synonymous predicates. We focus on optimizing predicates that do not involve merging and comparing their performance to demonstrate the effectiveness of our approach.

We experimentally compare the performances of the same reasoning model on the original KG and the optimized version obtained through predicate decomposition. We utilize clustering quality metrics to adaptively search for the optimal cluster that leads to optimal predicate decomposition, further enhancing the reasoning model's performance on KGs.

Overall, our work addresses crucial issues in predicate set configuration and provides a practical solution to enhance the quality of KGs and improve the performance of reasoning models. We introduce a versatile and practical method for improving KG reasoning performance through predicate decomposition. This approach is applicable to various KGs and reasoning models. We pioneer the implementation of predicate inductive reasoning that does not rely on external information and enables authentic KG completion tasks even when the original predicate set varies. This method also serves as a tool to enhance KG quality. We have optimized common KG benchmarks using this method and made the optimized versions publicly available.

The structure of this article is as follows: Section 2 mainly describes related works, such as KGC, inductive KGC, and predicate inductive KGC. Section 3 formalizes the methodology of the PDEC framework and provides preliminary theoretical proof. Section 4 describes the experimental datasets, baselines, and experimental setup details, and presents the experimental results. Section 5 discusses and analyzes the experimental results. Section 6 summarizes the work of the paper and analyzes the current limitations and future work plans.

## 2. Related Work

The work related to this article encompasses three primary facets: transductive KGC, inductive KGC, and predicate inductive KGC. Transductive KGC represents a conventional reasoning task, with trained KGC models unable to engage in reasoning related to novel entities absent from the training set. By contrast, inductive KGC allows the reasoning model to handle new entities not seen during training, but falls short when dealing with novel predicates. Predicate inductive KGC demands that the reasoning model possess the capability to reason over both new entities and predicates. Our PDEC framework executes the predicate inductive KGC task, which involves decomposing polysemous predicates within the original predicate set to yield new, more rational predicates.

As these three KGC tasks are progressively related and strongly correlated with PDEC, it becomes imperative for us to examine their associated literature. Compared with transductive KGC and inductive KGC, PDEC has performance advantages and unique capabilities in handling new predicates. Unlike existing models in the predicate inductive KGC realm, PDEC does not necessitate external information, instead solely relying on the analysis of the polysemy in the original predicate set to acquire reasoning capabilities for novel predicates. This approach, which is rooted in predicate decomposition, offers a more rational predicate set, thereby effectively enhancing the reasoning models' performance.

**Transductive KGC via KG embeddings.** A number of works are proposed for KGC tasks using learnable embeddings for KG relations and entities. For example, in [1–4,13], they learn to map KG relations into vector space and predict links with scoring functions. NTN [14], on the other hand, parameterizes each relation into a neural network. In [15], the authors present a theoretical framework that highlights the capabilities of graph neural networks (GNNs) in embedding entities and relations within KGs and executing link prediction tasks. The paper [16] proposes a divide–search–combine algorithm, RelEns-DSC, to efficiently search relation-aware ensemble weights for KG embedding. Because these algorithms need to learn embeddings of entities and relations in the test set during the training process, they are generally only suitable for transductive KGC tasks and cannot be applied to scenarios where there are new predicates in the test set such as predicate decomposition.

**Inductive KGC**. In recent years, inductive KG reasoning has gained increasing attention. This approach enables reasoning tasks to be performed in a bottom–up manner, focusing on the emergence of new entities. Methods such as ([17,18]) have emphasized the importance of modeling emerging entities, while ([19]) has introduced rule-based attention weights and ([20]) has extended RotatE ([4]) to enhance inductive reasoning. Alternatively, some research has focused on conducting inductive KGC tasks through rule mining. Neural LP [7] and DRUM [8] have reduced the rule learning problem to algebraic operations on neural-embedding-based representations of a given KG. LogCo [21] combines logical reasoning with contrastive representations and extracts subgraphs and relational paths to achieve entity independence and addresses supervision deficiencies, achieving superior performance on inductive KG reasoning. The paper [22] proposes an adaptive propagation path learning method for GNN-based inductive KG reasoning, addressing the limitations of hand-designed paths and explosive growth of entities. Compared with these methods, we tackle a more challenging problem where predicates can be new instead of only some entities being new.

**Predicate inductive KGC**. Predicate induction KGC is the latest development of inductive KGC (also known as entity induction KGC), which aims to enable KG reasoning models to perform KGC tasks on new predicate sets that do not exist in the training set. INGRAM [11] exhibits remarkable reasoning capabilities for novel predicates, enabling it to handle any new predicates. However, novel predicates are not generated via polysemy splitting and synonymy merging; furthermore, a message-passing step on the graph containing novel predicates is still required prior to reasoning. This makes it incapable of effectively detecting missing predicates. Although RMPI [11] exhibits some ability to discover new predicates, it relies on additional ontology information. These characteristics hinder efforts to optimize the quality of the original graph and the performance of reasoning models built upon it, facing challenges in enhancing reasoning abilities. In contrast, PDEC excels at

discovering new predicates by decomposing polysemous predicates and enhancing the reasoning performance of the original KGs.

## 3. Materials and Methods

To address the issue of determining the granularity of predicate decomposition, the PDEC framework is based on iterative optimization. A key feature of this framework is an automated predicate decomposition algorithm that identifies the predicates that require decomposition and determines their appropriate decomposition granularity without requiring human intervention.

In each iteration of the decomposition algorithm, entity representations are utilized to decompose predicates, as shown in Figure 1. This process updates the KG and then updates the entity representations using any representation learning algorithm. The updated entity representations serve as input for the next round of predicate decomposition.

Additionally, we present an optimization framework that allows control over the number of iterative optimization rounds with only a limited set of hyperparameters. Algorithm 1 provides a pseudocode overview of the PDEC algorithm.

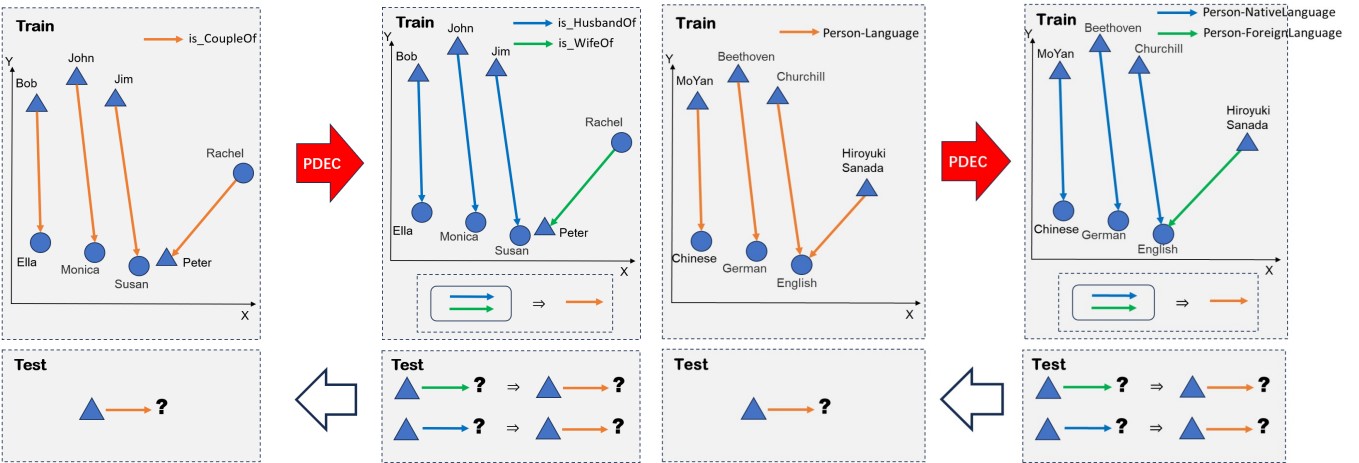

(**a**) Example of decomposing for predicate `is_CoupleOf` .   (**b**) Example of decomposing for predicate `Person-Language`.

**Figure 1.** The figure is a visual representation of the predicate decomposition framework. The left (**a**) is an example of the predicate `is_CoupleOf`. The circles and triangles represent entities, and the vectors between them represent predicates. Bob and Ella, John and Monica, Jim and Susan, and Rachel and Peter are all couple pairs; however, the relationship between Rachel and Peter is distinctly different from those of the other three pairs. This difference is emphasized by the orange lines in the figure. After predicate decomposition, the predicates can be divided into two separate predicates: `is_husbandOf` and `is_wifeOf`. This division is clearly demonstrated in the right figure. On the right (**b**) is another example of the predicate `Person-Language`. Please refer to the introduction for relevant explanations.

### 3.1. Preliminaries

**KGs**. A KG is a collection of the triples $\mathcal{K} = \{(h,r,t)|h,t \in \mathcal{E}, r \in \mathcal{R}\} \subseteq \mathcal{E} \times \mathcal{R} \times \mathcal{E}$, where $\mathcal{E}$ and $\mathcal{R}$ represent the sets of entities and predicates in the KG, respectively. In a triple $(h,r,t)$, entities $h$ and $t$ are referred to as the *head entity* and *tail entity*, respectively.

**KG embedding and notations**. Within the context of a KG $\mathcal{K} = \{(h,r,t)\}$, KG embedding aims to represent each entity $h, t \in \mathcal{E}$ and relation $r \in \mathcal{R}$ using continuous vectors, denoted as $\mathbf{h}, \mathbf{t}$, and $\mathbf{r}$, respectively. These vectors serve as dense representations that capture the meaning and relationships encoded in the KG.

**KGC tasks**. KGC, or knowledge graph completion, involves inferring missing facts from the known facts within KGs. The objective of a KG reasoning model is to effectively rank positive triplets higher than negative triplets, thereby accurately identifying potential positive triplets that may have been overlooked in the current graph.

Formally speaking, KGC is often cast as a *tail prediction* task, where the model aims to prioritize the tail entity $t \in \mathcal{E}$ of an incomplete triplet $x = (h, r, ?)$ over a set of negative entities. This set of negative entities is defined as $N_t = \{e \in \mathcal{E} : (h, r, e) \notin \mathcal{K}\}$, where $\mathcal{K}$ represents the known facts in the KG. The model, denoted as $F(x)$, computes a score vector $s$ for each entity $e \in \{t\} \cup N_t$. These scores indicate the likelihood of the triplet $(h, r, e)$ being true. Analogously, the *head prediction* task strives to complete incomplete triplets like $x = (?, r, t)$.

*3.2. Predicate Decomposition*

In this step, we aim to decompose the predicates in order to better represent contextual information. In existing representation learning frameworks, the same predicate is associated with the same representation vector, which cannot be further decomposed.

To discriminate polysemous predicates, we obtain the necessary contextual information for each predicate through its corresponding *edge vectors*. Edge vectors of $r_i$ can be formalized as follows:

$$\mathbf{e}_i = f_{\text{edge}}(\mathbf{h}_i, \mathbf{t}_i), \quad \forall (h_i, r_i, t_i) \in \mathcal{K}_{train} \cup \mathcal{K}_{valid} \tag{1}$$

For the sake of computational efficiency, we first employ efficient algorithms such as transE to learn the embeddings of entities and predicates in KGs. This allows us to obtain an initial model $\Psi$. We then utilize this initial model $\Psi$ to calculate edge vectors:

$$\vec{tr}_{a,i,j} = edge(\vec{t}_j, \vec{h}_i) \quad \forall (h_i, r_a, t_j) \in \mathcal{K}_{train} \cup \mathcal{K}_{valid} \tag{2}$$

where $\vec{h}$ and $\vec{t}$ represent the embedding vectors of $h$ and $t$ obtained by the representation learning model, while *edge* is a function that utilizes the mechanism of the model $\Psi$ to estimate the embedding of $r_a$ based on $\vec{h}$ and $\vec{t}$.

For instance, in TransE, Equation (3) specifies how this estimation is achieved:

$$\vec{tr}_{a,i,j} = \vec{t}_j - \vec{h}_i, \quad \forall (h_i, r_a, t_j) \in \mathcal{K}_{train} \cup \mathcal{K}_{valid} \tag{3}$$

Let $\nabla = \{\vec{tr} \in \mathbb{R}^d | \forall (h, r, t) \in \mathcal{K}_{train} \cup \mathcal{K}_{valid}\}$ represent the set of edge vectors. For each predicate $r_a \in \mathcal{R}$, its edge vector subset is $\nabla_a = \{\vec{tr}_{a,i,j} \in \mathbb{R}^d | \forall (h_i, r_a, t_j) \in \mathcal{K}_{train} \cup \mathcal{K}_{valid}\}$. We perform a clustering on the vector set $\nabla_a$ and use the Calinski–Harabasz index [23] as the clustering quality evaluation method. The clustering result is denoted as $\mathbf{CL}_{a,\theta} = \{\nabla_{a,0}, \nabla_{a,1}, \ldots \nabla_{a,K-1} \subset \nabla_a\}$, where $\theta$ is the parameter of the clustering algorithm, and $K$ is the number of clusters.

The optimal clustering result is determined according to the following equation:

$$\arg\max_{\theta} c\_h(\mathbf{CL}_{a,\theta}) \tag{4}$$

where $c\_h(\mathbf{CL}_{a,\theta})$ represents the Calinski–Harabasz index of the clustering result $\mathbf{CL}_{a,\theta}$.

Let the lower threshold of clustering quality be $\eta$. For every $r_a \in \mathcal{R}$, if $c\_h(\mathbf{CL}_{a,\theta}) > \eta$, we believe that the predicate $r_a$ is a polysemous predicate and can be split into $K$ new predicates $r_{a,0}, \ldots, r_{a,K-1}$; i.e., we perform the following predicate splitting:

$$(h_i, r_a, t_j) \rightarrow (h_i, r_{a,x}, t_j), \quad if \quad \vec{tr}_{a,i,j} \in \nabla_{a,x} \&\& c\_h(\mathbf{CL}_{a,\theta}) > \eta \tag{5}$$

where $x \in \mathbb{N}, x < K$. Let $split_a = \{r_{a,0}, \ldots, r_{a,K-1}\}$ be the set of predicates formed by splitting the predicate $r_a$. We denote the set of polysemous predicates as $\mathcal{R}_P$.

Given that the Calinski–Harabasz index often reaches its maximum when the number of clusters is 2 or 3, and that the semantic diversity of polysemous predicates typically does not exceed 3, to simplify the computational complexity, we utilize clustering algorithms that allow for the specification of the number of clusters, such as K-means, with $K = 2, 3$.

### 3.3. KG Reasoning Based on Predicate Decomposition

We perform the splitting of polysemous predicates through Equation (5), and obtain new datasets $\mathcal{K}'_{train}, \mathcal{K}'_{valid}$, which have $N'$ predicates. It is easy to know that $N' = |\mathcal{R}| + \sum_{r_a \in \mathcal{R}_P}(|split_a| - 1)$.

We use $\mathcal{K}'_{train}$ and $\mathcal{K}'_{valid}$ as training and validation sets to train the KG reasoning model. We adopt the same training method as the original paper of the baseline model to verify the effectiveness of the PDEC framework. For example, for the TransE model, minimize the following hinge loss:

$$\mathcal{L} = \sum_{(h,r,t) \in \mathcal{K}'_{train}} \sum_{(h',r,t') \in \bar{\mathcal{K}}'_{train}} \left[\gamma + Dist(\vec{h} + \vec{r}, \vec{t}) - Dist(\vec{h'} + \vec{r}, \vec{t'})\right]_+ \tag{6}$$

where $Dist(\vec{h} + \vec{r}, \vec{t})$ denotes the dissimilarity measure, which we take to be either the $L_1$ or the $L_2$-norm; $[x]_+$ denotes the positive part of $x$; $\gamma > 0$ is a margin hyperparameter; and $\bar{\mathcal{K}}'_{train}$ is the negative sample training set generated from $\mathcal{K}'_{train}$ in a 3:1 ratio as follows:

$$\bar{\mathcal{K}}'_{train} = \left\{ \begin{aligned} &(h',r,t')|(h,r,t) \in \mathcal{K}'_{train}, \\ &\left(h',r,t'\right) \notin \mathcal{K}'_{train}, h' \neq h \vee t' \neq t \end{aligned} \right\} \tag{7}$$

Let the trained model be $\Theta$. When using the model $\Theta$ to perform KGC tasks on the test set $\mathcal{K}_{test}$, follow the rules:

$$\Theta_k(h_i, r_a, t_j) = \bigvee_{r_{a,x} \in split_a} \Theta_k(h_i, r_{a,x}, t_j) \tag{8}$$

where $\Theta_k(h_i, r_a, t_j)$ is a logical function, representing the judgment of the model $\Theta$ on whether triplets $t_j$ is the top $k$ prediction of the query $(h_i, r_a, ?)$, or whether triplets $h_i$ is the top $k$ prediction of the query $(?, r_a, t_j)$.

We continue to use the example of the predicate `Person-Language` to explain the reason for this approach. Assume that PDEC decomposes `Person-Language` into two predicates, `Person-NativeLanguage` and `Person-ForeignLanguage`, representing the relationships between persons and their native languages and other languages, respectively. If the model $\theta$ determines that (A, `Person-NativeLanguage`, `English`) is reasonable (indicating that the native language of person `A` is `English`), then in all cases, `English` must be the language spoken by `A`. Therefore, (A, `Person-Language`, `English`) is reasonable. Similarly, if (A, `Person-ForeignLanguage`, `English`) is reasonable, then (A, `Person-Language`, `English`) is also reasonable. In essence, the new predicates induced by PDEC are subordinate to the original predicate. This design enables PDEC to achieve performance improvement in predicting links on KGs.

### 3.4. The Adaptive Optimization Mechanism of PDEC

We use the entity embeddings obtained from the model $\Theta$ to update the edge vector set $\nabla$ using Equation (2), and then regeneralize the polysemous predicates according to Equation (5), thereby achieving iterative optimization.

To adaptively determine $\eta$, for each iteration round, we order the Calinski–Harabasz index of the clustering results $CL_\theta$, and determine the clustering quality threshold $\eta$ according to the following rule:

$$\frac{\sum_{c\_h(\mathbf{CL}_{a,\theta}) > \eta, r_a \in \mathcal{R}} (1)}{\sum_{c\_h(\mathbf{CL}_{a,\theta}) <= \eta, r_a \in \mathcal{R}} (1)} < 0.8 \tag{9}$$

That is, we perform predicate decomposition on the predicates that have an index size in the top 80%. An empirical optimal hyperparameter of 0.8 is obtained from experiments, and the relevant experimental results are presented in Figures 7–9.

Subsequently, we use $\mathcal{K}'_{train}$ and $\mathcal{K}'_{valid}$ to retrain $\Theta$ and complete an iteration round. After each round, we assess the changes in mean reciprocal rank (MRR). If the increase in MRR is less than the early-stopping threshold, we terminate the iteration.

The algorithm process is outlined in Algorithm 1.

---

**Algorithm 1** KG reasoning based on polysemous predicate induction.

---

**Require:** Training set $\mathcal{K}_{train}$, validation set $\mathcal{K}_{valid}$, test set $\mathcal{K}_{test}$, threshold of clustering quality $\eta$, threshold of early stopping $\lambda$

**Ensure:** The reasoning model $\Theta$, trained using $\mathcal{K}'_{train}$ and $\mathcal{K}'_{valid}$ based on polysemous predicate induction. MRR and other indicators obtained from testing on $\mathcal{K}_{test}$.

1: Train the initial model $\Psi$ based on the original KG $\mathcal{K}$.
2: Using $\Psi$, calculate the entity embedding set $\vec{E} = \{\vec{e}|e \in \mathcal{E}\}$ and predicate embedding set $\vec{R} = \{\vec{r}|r \in \mathcal{R}\}$.
3: **for** $t \leftarrow 1$ to $T$ **do**
4:    Based on $\vec{E}$, update the edge vector set $\nabla$ using Equation (2);
5:    **for** $a \leftarrow 0$ to $|\mathcal{R}|$ **do**
6:       Perform clustering on the edge vector set $\nabla_a$ according to Equation (4);
7:       **if** $c\_h(\mathbf{CL}_{a,\theta}) > \eta$ **then**
8:          update $split_a$ using Equation (5);
9:       **end if**
10:    **end for**
11: **end for**
12: Construct/update $\mathcal{K}'_{train}$ and $\mathcal{K}'_{valid}$ according to $split_a, a \in \mathcal{R}$;
13: Train the reasoning model $\Theta$ based on $\mathcal{K}'_{train}$ and $\mathcal{K}'_{valid}$;
14: Based on $\mathcal{K}_{test}$, use Equation (8) to calculate the performance of the model $\Theta$ and record the results as $M^t = MRR^t, HIT1^t, \ldots$
15: udpate $\vec{E}$
16: **if** $t == 0 || MRR^t - MRR^{t-1} > \lambda$ **then**
17:    Continue
18: **end if**

---

### 3.5. Synonymous Predicate Merging

To further investigate the impact of predicate semantics on KG reasoning tasks, we also conducted experiments on synonymous predicate merging. We clustered the predicate embeddings generated by the initial model $\Psi$. By adjusting the number of clusters, we could control the degree of synonymous predicate merging. For example, on the FB15K-237 dataset, if we set the number of predicate clusters to 200, a maximum of 37 predicates are considered synonymous. When the number of clusters is 230, the maximum number of synonymous predicates is 7.

After merging synonymous predicates, the trained model focuses on the hypernyms of the original predicates, making it impossible to perform KG completion tasks related to the predicates involved in the merging process. However, it is still possible to perform KG completion tasks related to predicates that were not merged. Please refer to Section 4.2 for detailed experimental results.

### 3.6. Theoretical Proof of PDEC

We conducted a theoretical analysis to demonstrate the effectiveness of PDEC in the TransE model. Consider the toy KG $K_t \subseteq \mathcal{E}_t \times \mathcal{R}_t \times \mathcal{E}_t$ represented in Figure 2. If we use TransE for learning entity embeddings, the optimization objective would be as follows:

$$\arg\max_\theta \|x + r_0 - a\| + \|x + r_0 - b\| + \|x + r_0 - c\| + \|x + r_0 - d\| \tag{10}$$

where $x, a, b, c, d \in \mathcal{E}_t$ are entities in the KG, and $(x, r_0, a), (x, r_0, b), (x, r_0, c), (x, r_0, d) \in K_t$. The predicate $r_0$ is represented by an orange line in Figure 2.

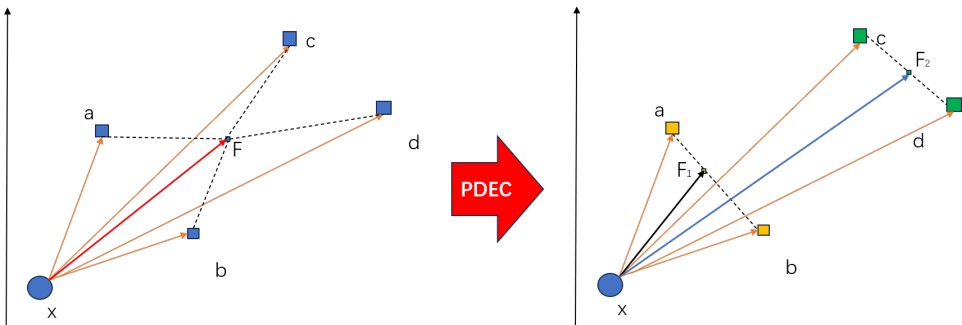

**Figure 2.** Schematic diagram of the predicate decomposition principle. The theoretical error of TransE algorithm optimization can be approximately estimated using the length of dashed lines.

It is easy to observe that after optimization, the representation of the predicate $r_0$ approaches the vector from $x$ to the Fermat point $F$ of the quadrilateral formed by $a, b, c$, and $d$. The deviation $dev_0$ is equivalent to the sum of distances from $F$ to $a, b, c$, and $d$, that is, $dev_0 = |Fa| + |Fb| + |Fc| + |Fd|$.

After PDEC processing, the edge vectors $\vec{tr}_{0,x,a}$ and $\vec{tr}_{0,x,b}$ form one cluster, while $\vec{tr}_{0,x,c}$ and $\vec{tr}_{0,x,d}$ form another cluster, thus decomposing the predicate $r_0$ into $r_1$ and $r_2$. This means that the triples $(x, r_1, a), (x, r_1, b), (x, r_2, c), (x, r_2, d) \in K_t$. Consequently, the optimization objective transforms to the following:

$$\arg\max_\theta \|x + r_1 - a\| + \|x + r_1 - b\| \|x + r_2 - c\| + \|x + r_2 - d\| \tag{11}$$

where the final optimized predicate $r_1$ is actually approaching the vector from $x$ to the Fermat point $F_1$ of the line formed by $a, b$, and $F_1$ must be on the line $\vec{ab}$. Therefore, the loss is equivalent to the sum of the distances from $F_1$ to $a$ and $b$, i.e., $dev_1 = |F_1 a| + |F_1 b|$. Similarly, the deviation of embedding learning for $r_2$ is $dev_2 = |F_2 c| + |F_2 d|$, and $F_2$ must be on the line $\vec{cd}$. As shown in Figure 3, according to the geometric properties of triangles, it can be seen as follows:

$$\begin{aligned} dev_1 + dev_2 &= |F_1 a| + |F_1 b| + |F_2 c| + |F_2 d| \\ &< |Fa| + |Fb| + |Fc| + |Fd| = dev_0 \end{aligned} \tag{12}$$

The theoretical analysis of the toy KG reveals that in the case where the head entity of the triplet associated with the predicate $r$ in KG is unique, TransE's optimization process aims to find the Fermat points $F$ for all tail entities. The obtained embedding of $r$ corresponds to the vector between the head entity and the Fermat point $F$, with the error measured as the distance between $F$ and all tail entities.

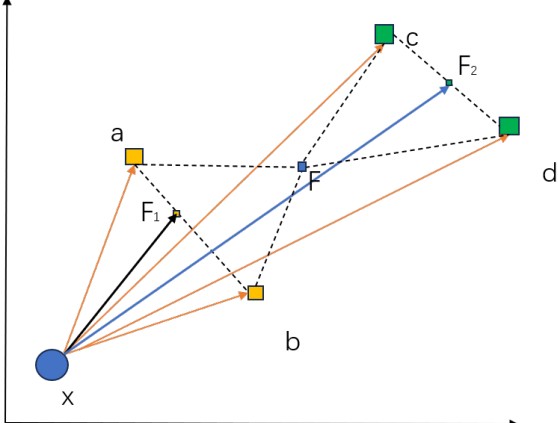

**Figure 3.** Advantages of predicate decomposition. Based on the geometric characteristics of triangles, PDEC reduces the theoretical error of TransE algorithm optimization (represented by dashed lines).

By clustering edge vectors, predicate decomposition transforms the task of finding Fermat points for all global tail entities into finding local Fermat points for several clusters. The error is now the sum of distances from the Fermat points of each cluster to their corresponding tail entities. This approach inevitably reduces the total error and improves TransE's performance.

When there are multiple head entities for the predicate *r*, the analysis can be applied to each individual head entity. In this case, the embedding of the original predicate obtained by TransE corresponds to the vector between Fermat points of clusters formed by head entities and Fermat points of clusters formed by tail entities. This observation further supports the claim that PDEC reduces the error in the TransE model even when dealing with multiple head entities.

## 4. Results

We experimentally verified that PDEC can reasonably judge polysemous predicates and significantly improve the performance of reasoning models on the KG. Usually, we test the reasoning ability of a model by performing KGC tasks.

### 4.1. Experimental Setup

**Datasets**. Open-world KGC tasks are commonly evaluated on Word-Net and Freebase subsets, such as YAGO3 [24] and FB15K-237 [25]. In order to verify the effectiveness of predicate decomposition, we focus on the KG benchmark with many predicates and high difficulty to verify the effectiveness of our method. Therefore, we selected FB15K-237, YAGO3-10, and NELL-995 [26] as the benchmark dataset.

- FB15K-237 is a subset of the Freebase knowledge base [10] containing general knowledge facts.
- The YAGO3-10 dataset is a subset of YAGO3 that only contains entities with at least 10 relations. In total, YAGO3-10 has 123,182 entities and 37 relations and 1,179,040 triples, and most of the triples describe attributes of persons such as citizenship, gender, and profession.
- The NELL-995 dataset is a subset of NELL [27] created from the 995th iteration of the construction. NELL-995 includes 75,492 entities, 200 relations, and 154,208 triples.

To verify the efficacy of this method for knowledge extraction and logical reasoning on large-scale datasets, we also conducted drug rediscovery experiments on the open-source biochemical knowledge graph RTX-KG2c [28]. RTX-KG2c integrates data from 70 public knowledge sources into a comprehensive graph where all biological entities (e.g., "ibuprofen") are represented as nodes and all concept–predicate–concept relationships (e.g., "ibuprofen increases activity of GP1BA gene") are encoded as edges. This dataset comprises approximately 6.4 M entities across 56 distinct categories, with 39.3 M relationship edges described by 77 distinct relations. The objective of this experiment is to employ the KGC model to learn the interactions between diseases and drugs from RTX-KG2c, aiming to predict potential therapeutic relationships between drugs and diseases.

**Baselines**. In order to test the effectiveness and universality of PDEC, we have extensively selected some mature KG inference models. We use the KG inference model running on the original dataset as the baseline. For each baseline, we use them in conjunction with PDEC to conduct performance testing and record their performance improvement. The baseline model we have chosen includes TransE, TransH [29], TransR [30], TransD [31], ComplEX [32], DistMult [2], TuckER [13], RotatE, CompGCN [33], RelEns-DSC [16], etc.

**Experiment setting details**. We implement the baseline model and its corresponding PDEC framework based on the OPENKE project [34]. We set the entity and relation embedding size to 200 for experiments. We use Adam optimization [35] and search the learning rate (0.001–0.005) and minibatch size (64–256). We apply dropout to the entity and relation embeddings and all feed-forward layers, and search the dropout rates within 0.6.

In line with the common practices described in [5,7], we compute standard evaluation metrics for the KGC reasoning task, *Hit@k*, which counts the number of correctly predicted

head terms among the top $k$ predictions, and the mean reciprocal rank (MRR), calculated as the mean of the reciprocal rank of the correct answer in the list of predictions. These ranking-based metrics are defined as follows:

$$MPR = \frac{1}{R} \sum_{r \in R} r^{-1} \quad Hit@k = \frac{1}{R} \sum_{r \in R} |r \leq k| \tag{13}$$

where $R$ is a list of ranks of all true-positive triples in the test dataset.

For drug rediscovery experiments, we excluded all existing edges connecting potential drug nodes (nodes labeled "Drug" or "SmallMolecule") with potential disease nodes (nodes labeled "Disease", "PhenotypicFeature", "BehavioralFeature", or "DiseaseOrPhenotypicFeature") in RTX-KG2c to prevent information leakage during training. We then added drug–disease pairs that were confirmed true positives (pairs with the relation "indication" from MyChem datasets [36] or the predicate "treats" from SemMedDB datasets [37]). A new predicate `treat` was introduced to represent this therapeutic relationship in the experimental KG. We generated new triples based on these positive drug–disease pairs and added them to the KG, dividing them into training, validation, and testing sets in a 7:2:1 ratio. In the experiment, we only performed KGC tasks on the triples related to the newly added predicate `treat` in the test set to verify the drug rediscovery ability of the model.

To implement our approach, we utilized the PyTorch library in Python. All experiments were conducted on a machine equipped with 6 Nvidia Tesla V100 GPUs and 32 GB RAM.

### 4.2. Experiment Results

The experimental results demonstrate that PDEC possesses the capability to perform predicate inductive KGC tasks through predicate decomposition, and the original predicate set can be effectively restored using Equation (8), leading to an enhancement in the reasoning performance for the original transductive KGC task. In the subsequent sections, we will present comprehensive experimental results showcasing PDEC's impact on improving the reasoning performance of the original KGC task. These results cover various benchmarks and large datasets, i.e., drug rediscovery. Furthermore, we will demonstrate the correlation between the granularity of predicate decomposition and the corresponding performance improvements. Lastly, we will evaluate the influence of synonymous predicate merging on the overall performance of the KGC task.

**The KGC performance improvement on benchmarks brought by PDEC**. We tested the performance of the baseline reasoning model on FB15K-237, YAGO3-10, and NELL-995, as well as the performance improvement of PDEC, as shown in the Table 1. The initial edge vectors are generated using TransE. We performed PDEC using Algorithm 1, resulting in datasets with new predicates. All tests ensure that the reasoning model keeps hyperparameters unchanged during the testing of the original benchmark and the benchmark after PDEC to ensure a fair comparison. The results indicate that PDEC can effectively promote the performance of KG reasoning models.

**Table 1.** The performance gain of PDEC on baseline models in benchmark experiments. Hits@k is in %.

| Method | PDEC | FB15K-237 | | | | YAGO3-10 | | | | NELL-995 | | | |
|---|---|---|---|---|---|---|---|---|---|---|---|---|---|
| | | MRR | Hits@1 | Hits@3 | Hits@10 | MRR | Hits@1 | Hits@3 | Hits@10 | MRR | Hits@1 | Hits@3 | Hits@10 |
| TransE | ✘ | 0.289 | 19.3 | 32.6 | 47.9 | 0.330 | 21.5 | 38.8 | 54.9 | 0.252 | 9.64 | 38.5 | 47.2 |
| | ✔ | 0.349 | 24.2 | 39.9 | 55.8 | 0.395 | 28.0 | 46.0 | 60.3 | 0.287 | 11.7 | 43.0 | 52.7 |
| DistMult | ✘ | 0.187 | 10.3 | 20.7 | 36.1 | 0.073 | 2.93 | 6.87 | 16.7 | 0.163 | 7.10 | 19.1 | 33.9 |
| | ✔ | 0.224 | 13.3 | 25.4 | 40.9 | 0.112 | 3.93 | 12.0 | 28.5 | 0.186 | 9.0 | 21.6 | 35.4 |
| TransH | ✘ | 0.286 | 18.4 | 32.9 | 48.4 | 0.332 | 22.1 | 38.9 | 54.4 | 0.255 | 10.0 | 39.6 | 48.8 |
| | ✔ | 0.314 | 20.6 | 36.1 | 52.2 | 0.385 | 26.6 | 43.7 | 62.2 | 0.278 | 13.5 | 40.7 | 51.6 |
| RotatE | ✘ | 0.321 | 22.8 | 35.6 | 50.6 | 0.270 | 17.9 | 30.3 | 44.6 | 0.368 | 31.4 | 40.5 | 45.6 |
| | ✔ | 0.360 | 26.4 | 40.2 | 54.7 | 0.355 | 25.6 | 39.8 | 54.3 | 0.374 | 32.1 | 41.1 | 46.0 |
| TransR | ✘ | 0.305 | 20.8 | 34.4 | 49.6 | 0.321 | 11.6 | 48.6 | 63.7 | 0.261 | 10.8 | 39.4 | 49.0 |
| | ✔ | 0.334 | 23.5 | 37.8 | 53.3 | 0.496 | 39.2 | 56.0 | 68.3 | 0.276 | 11.7 | 41.0 | 53.3 |
| TransD | ✘ | 0.284 | 18.1 | 32.7 | 48.6 | 0.323 | 21.5 | 35.9 | 53.8 | 0.263 | 10.5 | 39.9 | 50.1 |
| | ✔ | 0.310 | 20.0 | 36.1 | 51.9 | 0.377 | 25.1 | 42.6 | 61.7 | 0.280 | 12.1 | 40.4 | 50.8 |

**Table 1.** *Cont.*

| Method | PDEC | FB15K-237 | | | | YAGO3-10 | | | | NELL-995 | | | |
|---|---|---|---|---|---|---|---|---|---|---|---|---|---|
| | | MRR | Hits@1 | Hits@3 | Hits@10 | MRR | Hits@1 | Hits@3 | Hits@10 | MRR | Hits@1 | Hits@3 | Hits@10 |
| CompIEX | ✘ | 0.238 | 15.2 | 26.5 | 41.0 | 0.106 | 2.96 | 11.4 | 26.6 | 0.222 | 8.9 | 32.1 | 40.6 |
| | ✔ | 0.280 | 18.9 | 31.5 | 46.0 | 0.196 | 9.35 | 22.8 | 41.4 | 0.248 | 9.9 | 37.9 | 45.0 |
| TuckER | ✘ | 0.323 | 0.238 | 0.353 | 0.494 | 0.332 | 26.8 | 34.3 | 47.9 | 0.293 | 21.6 | 32.5 | 41.7 |
| | ✔ | 0.327 | 25.5 | 34.8 | 46.8 | 0.344 | 27.5 | 35.1 | 49.0 | 0.298 | 23.1 | 33.9 | 41.9 |
| CompGCN | ✘ | 0.355 | 26.4 | 39.0 | 53.5 | 0.411 | 37.9 | 48.3 | 57.4 | 0.461 | 38.0 | 49.1 | 58.9 |
| | ✔ | 0.363 | 26.9 | 40.3 | 54.6 | 0.428 | 38.3 | 49.9 | 58.8 | 0.481 | 39.2 | 51.1 | 60.1 |
| RelEns-DSC | ✘ | 0.368 | 27.4 | 40.5 | 55.6 | 0.342 | 27.9 | 36.1 | 49.1 | 0.548 | 48.2 | 59.0 | 66.7 |
| | ✔ | 0.377 | 28.6 | 43.1 | 56.9 | 0.349 | 27.8 | 35.7 | 49.6 | 0.562 | 50.0 | 61.2 | 67.5 |

We have drawn a bar chart to visually demonstrate the improvement of PDEC's KG reasoning ability through PDEC, as shown in Figures 4–6.

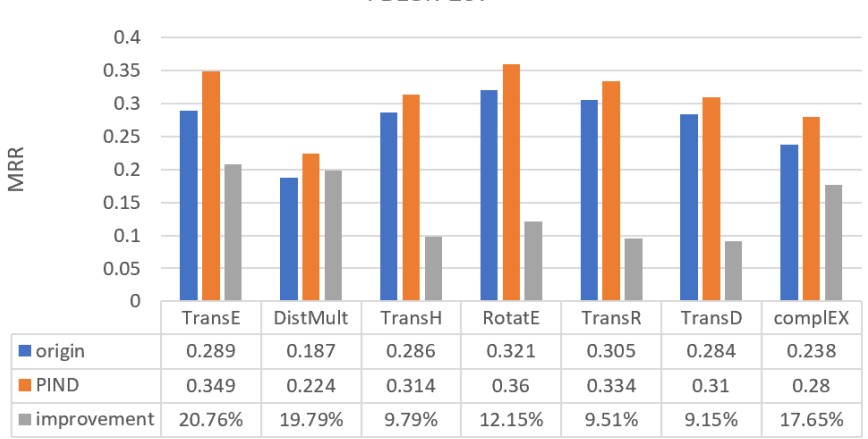

**Figure 4.** Performance improvement brought by PDEC to baseline on FB15K-237.

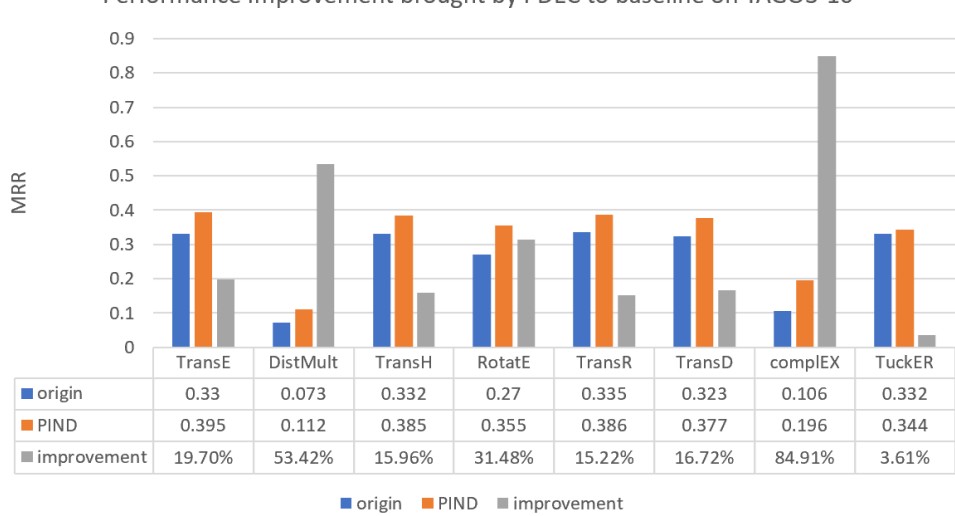

**Figure 5.** Performance improvement brought by PDEC to baseline on YAGO3-10.

**The KGC performance improvement brought by PDEC on a large-scale dataset (i.e., drug rediscovery).** We conducted KGC experiments on the large-scale biochemical knowledge graph RTX-KG2c and compared the performances of the baseline model before and after applying the PDEC framework. During the testing of both the original benchmark and the benchmark after PDEC, all tests guarantee that the reasoning model maintains consistent hyperparameters to ensure a fair comparison. The relevant results are shown in Table 2.

The results indicate that PDEC can effectively improve the ability of KG reasoning models on drug rediscovery.

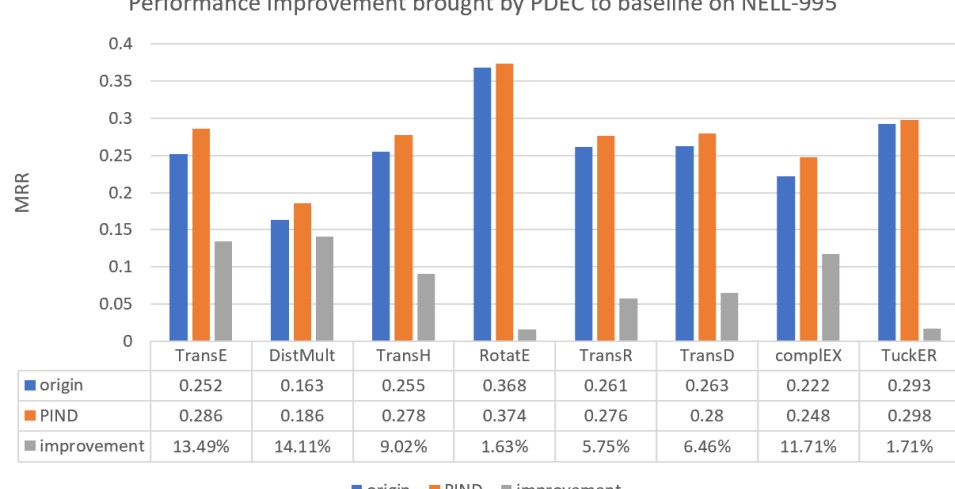

**Figure 6.** Performance improvement brought by PDEC to baseline on NELL-995.

**Table 2.** The performance gain of PDEC on baseline models in drug rediscovery experiments. Hits@k is in %.

| Method | PDEC | FB15K-237 | | | |
|---|---|---|---|---|---|
| | | **MRR** | **Hits@1** | **Hits@3** | **Hits@10** |
| TransE | ✘ | 0.232 | 17.3 | 45.0 | 58.1 |
| | ✔ | **0.269** | **20.3** | **48.1** | **61.1** |
| DistMult | ✘ | 0.164 | 16.0 | 30.3 | 34.8 |
| | ✔ | **0.177** | **17.1** | **33.4** | **35.9** |
| RotatE | ✘ | 0.296 | 22.8 | 49.1 | 57.9 |
| | ✔ | **0.302** | **23.4** | **49.2** | **57.7** |
| CompGCN | ✘ | 0.291 | 22.3 | 48.6 | 58.1 |
| | ✔ | **0.306** | **23.9** | **49.3** | **59.6** |

**The correlation between the granularity of predicate decomposition and KGC performance**. In order to investigate the impact of clustering result quality on PDEC performance, we tested the difference in PDEC's improvement to TransE's performance when the proportion of predicates applied for splitting was different. Thus, an experimental conclusion was obtained on the correlation between predicate decomposition granularity and KGC performance.

We adjust $\eta$ to split more predicates based on the clustering results. The experimental results indicate that when $\eta$ is greater than a certain degree, the size of $\eta$ is inversely proportional to the effect of PDEC, as shown in Figures 7–9. Usually, the optimal number of predicates after decomposition is around 180% of the original number of predicates. This indicates that when $\eta$ values are reasonable, the more clustering results are adopted for predicate splitting, the better is the PDEC effect.

**The impact of synonymous predicate merging on KGC performance**. We trained the dataset after predicate merging based on TransE and performed KGC tasks on the original predicates that did not participate in the merging. The maximum number of synonymous predicates was set to 0, 7, and 37. The results obtained are shown in Table 3.

**Table 3.** KGC results of TransE with predicates merging on the FB15K-237 dataset.

| Number of Predicates Merging | MRR of Predicate Merging Model on Unmerged Predicates | MRR of Original Model on Unmerged Predicates |
|---|---|---|
| 0 | 0.289 | 0.289 |
| 7 | 0.286 | 0.286 |
| 37 | 0.246 | 0.245 |

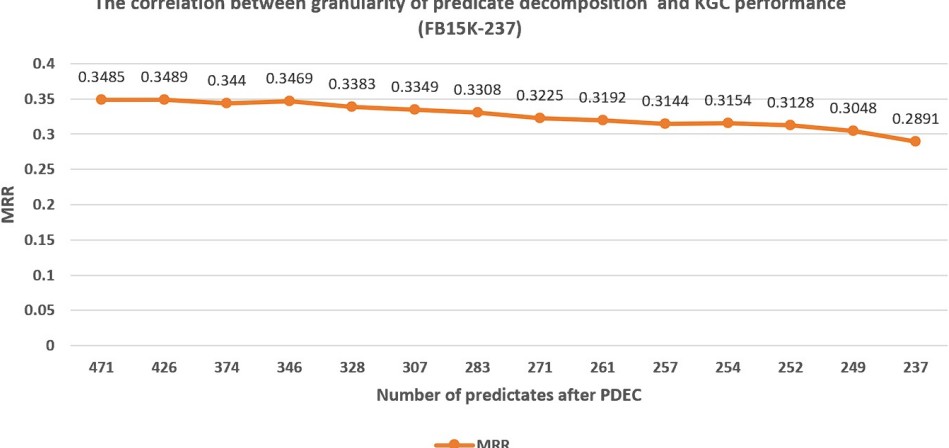

**Figure 7.** The correlation between granularity of predicate decomposition and KGC performance (FB15K-237).

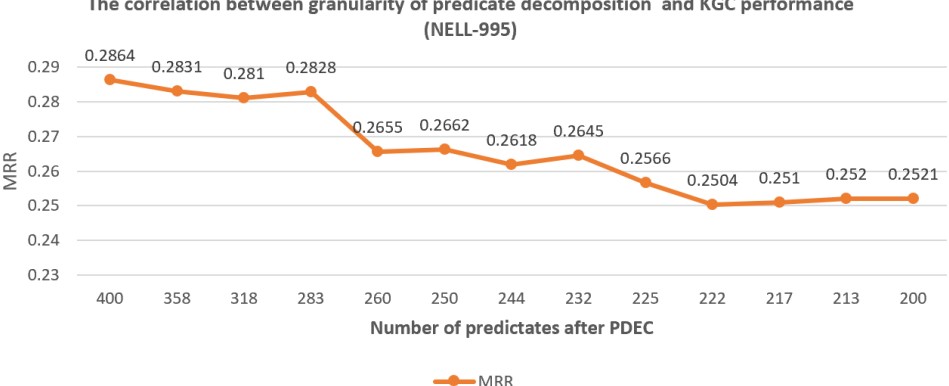

**Figure 8.** The correlation between granularity of predicate decomposition and KGC performance (NELL-995).

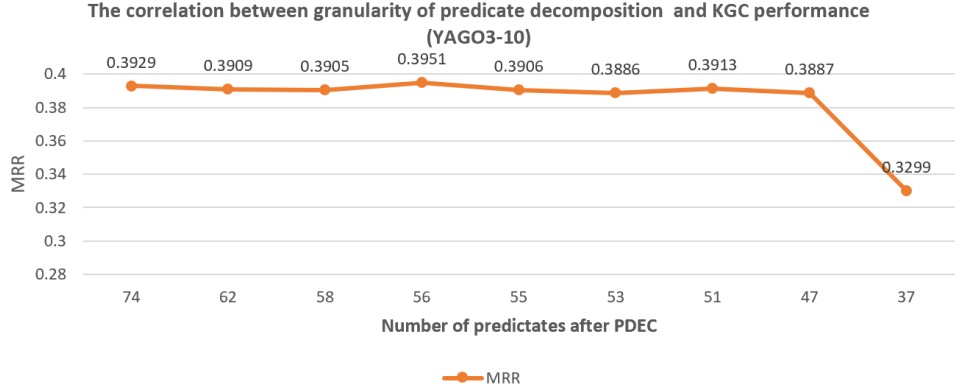

**Figure 9.** The correlation between granularity of predicate decomposition and KGC performance (YAGO3-10).

## 5. Discussion

### 5.1. The Performance Improvement of the Baseline Model after Applying the PDEC Framework

In Figure 4, it is evident that PDEC exhibits a notable enhancement in performance compared with traditional baseline methods on the FB15K-237 dataset. Notably, the TransE and DistMult methods achieve the most significant improvements, both achieving approximately 20% improvement.

As Figure 5 illustrates, PDEC demonstrates the most significant performance enhancement on the YAGO3-10 dataset when compared with traditional methods. Among the various methods, the CompIEX approach demonstrates the most remarkable improvement, achieving approximately 85% improvement, closely followed by DistMult with approximately 53% improvement.

On the NELL-995 dataset, Figure 6 reveals that PDEC demonstrates the least improvement over traditional methods. Nevertheless, the TransE and DistMult methods still exhibit the most notable improvements, achieving 13.5% and 14%, respectively.

As shown in Table 2, drug rediscovery experiments have shown that the PDEC framework can improve KG reasoning performance on larger datasets. Among them, TransE achieved the most significant performance improvement after adopting the PDEC framework, reaching 13.8%. In addition, PDEC achieved performance improvements of 7.9%, 2.0%, and 5.2% for DistMult, RotatE, and CompGCN, respectively.

### 5.2. The Characteristics of PDEC Framework in Performance Improvement

Based on the above experimental results, our experimental results lead to several key observations:

- PDEC generally enhances the reasoning performance of KGC's baseline model. When compared with the baseline methods, PDEC demonstrates consistent improvements across different datasets. This observation aligns with our objective of developing a more effective approach for KGC tasks.
- The performance improvement of PDEC is more significant on larger datasets. When examining the performance improvement of PDEC on different datasets, we observe that the larger the dataset, the more significant the improvement. This trend suggests that PDEC is particularly effective in capturing relationships and patterns present in larger KGs. It further supports our hypothesis that PDEC's ability to model polysemous predicates effectively enables it to handle the complexity and diversity found in larger datasets.
- The performance improvement of PDEC is more significant for old-fashioned methods. Our experiments also reveal that the performance improvement of PDEC is more significant for older methods like TransE and DistMult. These methods are known to be more susceptible to polysemous predicates, which are common in KGs. This observation aligns with our theoretical analysis presented in Section 3.6, where we discuss how PDEC addresses the limitations of traditional methods by effectively modeling polysemous predicates.

### 5.3. The Impact of Granularity of Predicate Decomposition on KGC Performance

As seen in Figures 7–9, we found that when the granularity of predicate decomposition is low, there is a roughly positive correlation between the granularity of predicate decomposition and the performance improvement of PDEC. When the clustering quality threshold $\eta$ is set high enough, the lower the parameter $\eta$ is set, the more new predicates are generated by decomposition, and the performance of PDEC improves accordingly.

We can also observe that PDEC performance reaches its optimal level when the number of new predicates generated by decomposition approaches the number of original predicates. However, as the number of new predicates generated by decomposition increases, the computational efficiency of PDEC will decrease, which needs to be balanced.

### 5.4. The Performance Impact of Synonymous Predicate Merging

The new predicate-related triples generated by synonymous predicate merging cannot correspond one-to-one to the triples in the original dataset, preventing direct KGC performance comparisons on these new triples. KGC performance improvement tests can only be conducted on the dataset corresponding to unmerged predicates.

According to the experimental results presented in Table 3, it can be seen that synonymous predicate merging has minimal performance improvement for triples that do

not involve merged predicates. Only a 0.001 improvement is observed when merging 37 predicates. This is because KGC is performed only on the dataset corresponding to unmerged predicates, leading to an indirect impact.

## 6. Conclusions

This article delves into the impact of predicate settings in KGs, aiming to enhance reasoning performance through predicate decomposition. We present PDEC, a technique that improves the performance of reasoning methods on KGs. This method is unique in that it implements predicate-inductive reasoning without introducing external information. When the original predicate set changes, PDEC can still perform KGC tasks effectively. Furthermore, this approach enhances the quality of KGs.

To validate the efficacy of PDEC, we conducted rigorous experiments. We analyzed the correlation between the granularity of predicate decomposition and the improvement in PDEC performance, as well as the impact of synonymous predicate merging. Our findings demonstrate the effectiveness of PDEC in enhancing reasoning performance on KGs.

**Limitations**. The limitations of this method mainly lie in the following: First, the automatic optimization of clustering threshold hyperparameters in Equation (9) has not been achieved yet. In fact, according to the discussion in Section 3.6, this hyperparameter can be estimated based on the low dimensional manifold distribution of the dataset. The current algorithm version's clustering threshold parameters are based on empirical values obtained from a large number of experiments, which affects the efficiency and practicality of the algorithm. We have conducted relevant research and exploration, but have not yet reached a clear conclusion. Second, PDEC's clustering-based predicate decomposition cannot fully correspond to the hypernyms and hyponyms of predicates in the ontology, which reduces the interpretability of the new predicate set after predicate decomposition. This is also the cost that PDEC attempts to avoid introducing external information. Third, it has not yet been possible to synchronize and optimize the decomposition of polysemous predicates with the merging of synonymous predicates.

Moving forward, our focus will be on automating the optimization of clustering threshold parameters for PDEC and combining ontology information to obtain new predicates that are more interpretable. We will also explore more effective optimization methods that balance polysemous predicate decomposition and synonymous predicate merging. These future directions aim to further improve the performance and quality of KGs.

**Author Contributions:** Conceptualization, X.T. and Y.M.; methodology, X.T. and Y.M.; software, X.T.; validation, X.T.; formal analysis, X.T.; investigation, X.T. and Y.M.; resources, X.T. and Y.M.; data curation, X.T.; writing—original draft preparation, X.T.; writing—review and editing, X.T.; visualization, X.T.; supervision, Y.M.; project administration, Y.M. All authors have read and agreed to the published version of the manuscript.

**Funding:** This research received no external funding.

**Data Availability Statement:** The FB15K-237 Knowledge Base Completion dataset is available at https://www.microsoft.com/en-us/download/details.aspx?id=52312 (accessed on 15 February 2024). The YAGO3-10 dataset can be downloaded at https://web.informatik.uni-mannheim.de/pi1/kge-datasets/yago3-10.tar.qz (accessed on 15 February 2024). The NELL-995 dataset can be downloaded at https://github.com/wenhuchen/KB-Reasoning-Data (accessed on 15 February 2024).

**Acknowledgments:** The work and writing of this thesis have received strong support and assistance from Xin Wang from the Department of Computer Science at Tsinghua University.

**Conflicts of Interest:** The authors declare no conflicts of interest.

## Abbreviations

The following abbreviations are used in this manuscript:

| | |
|---|---|
| KG | knowledge graph |
| KGC | knowledge graph completion |
| LLMs | large language models |
| GNNs | graph neural networks |

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
