# Peer review of "PDEC: A Framework for Improving Knowledge Graph Reasoning Performance through Predicate Decomposition"

_algorithms, doi:10.3390/a17030129_

Round 1

Reviewer 1 Report

Comments and Suggestions for Authors

The work developed by the authors is very pertinent and somewhat original and aims to create new methods for improving knowledge graphs reasoning performance in order to reduce ambiguity through the decomposition of predicates, thus reducing certain types of ambiguities related to polysemy. I consider, however, that the work lacks a certain balance, which causes it to present some flaws, which, once corrected, could make the article an important contribution to the state of the art in this area of Knowledge Graphs.

 I will now highlight some flaws and points that should be improved.

 1) When the authors say “We introduce a versatile and practical method for improving KG reasoning performance through predicate decomposition. This approach is applicable to various KGs and reasoning models”, it would be important to understand the degree of coverage and generalization of these methods for any KG. Are they only specific to some domains? Would it be possible to improve with the use of ontologies? It would be important for authors to include this type of information in the article.

 2) The Related Work section is a little poor and this is reflected in the small number of references used in the article. This section should be improved in order to more deeply encompass what is done upstream and downstream.

 3) It would be interesting to have one or two more examples of the application of PDEC to decompose predicates, in addition to the example in figure 1. This is to understand the type of predicates that the system can deal with. Eventually insert the list of predicates used by the system.

 The conclusions section should also be a little longer. You could possibly include information about the limitations of your approach, which could even answer question 1. You should also include what improvements you plan to make in the future.

 From a general point of view, the work could prove to be very useful for the scientific community and has merit on several levels, such as making the code available. Still, it requires essential review and improvement concerning the points mentioned above. For these reasons, I believe that the article should undergo a major revision for it to be published.

Reviewer 2 Report

Comments and Suggestions for Authors

This review report aims to give some pointers to the paper's authors entitled 'PDEC: a framework for improving knowledge graph reasoning performance through predicate decomposition". No doubt, the paper shows several strengths, such as how the authors address the subject matter in the introduction (lines 16-41) and spot the research gap (lines 46-61). Similarly, it is hard to fault how the authors preface the conclusion section with a summary (lines 351-361). Finally, the authors can win plaudits for putting forward future lines of research (lines 362-365).

However, the current version of the paper shows severe shortcomings as follows:

(1)   Although the authors set out the research objectives in the introduction, it could be more efficient and shorter (lines 61-86). Equally, the authors need to distinguish between general and specific objectives. Moreover, it is hard to identify the research target variables. Please indicate general and specific research objectives by indicating the research target variables.

(2) The paper contribution should be placed in the conclusion section rather than in the introduction section (lines 87-94). This is the conventional section to include this kind of content, and we should only count the chickens after they are hatched. Please move it to the conclusion section.

(3)   It is always advisable to append the paper structure description to the introduction. Please create a paragraph at the end of the introduction describing the paper's main headings and content.

(4)   Presumably, the literature heading is reviewed (lines 95-119). However, this is nothing more than three short messages without apparent interconnection. It needs to define the research target variables and articulate the variable's relationships. Additionally, I wonder if the authors might develop a doctrinal reasoning thread of ideas with theoretical propositions and hypotheses to test. I question the consistency between the 'related work' section and the analysis of the result section. I also need help finding the connection between the research objectives in the introduction and these triple definitions. Not only is it necessary to strengthen this essential theoretical section, but it is also crucial to build well-articulated and internally consistent threads of ideas. Please develop a proper literature review section.

(5)   If there is no inconsistency in the formula, experts might praise the authors for formulating algorithms for the predicate decomposition frameworks (lines 120-246). Nevertheless, the methodological section should include more than a formula development output. I would rather the authors include information about the universe and elements, sampling units and procedures, measuring instruments and statistical tools. If the authors use an experiment, they should explain the experimental design and describe the experimental subjects, external and internal variables, the treatments, etc. Otherwise, the reader has yet to see how all might be tested. Please include this methodological information systematically in the materials and methods section.

(6)   It is hard to get to grips with the analysis of the result section (lines 247-288) since there is no advancement of the section structure at the beginning. Similarly, the names of the statistical techniques remain dark, even though one might guess them. Moreover, when the author gives a name to the statistical technique, I question whether the term is correct. For example, it is said "impact", yet the statistical technique is a correlation (lines 298). Are they testing a causal effect and impact using a descriptive statistical technique?

(7)   As far as a discussion section is concerned, comparisons between other authors' results and ideas and the current paper's obtained evidence should be made to gain a much better understanding of the literature. Nevertheless, the discussion's current version has a summary (lines 313-330) and empirical reflections (lines 331-350) rather than the intention of delving into the obtained evidence by comparing other authors and their research work. To solve these shortcomings, several measures should be taken. First, move the summary to the conclusion section. Second, move the empirical reflection to the results section. Third, build a discussion between your obtained results and other papers' results to gain insight into your theoretical and empirical contribution.

(8)   Concluding is more than summarising and putting forward future lines of research (lines 351-365). Nevertheless, the current conclusion section needs more practical implications and limitations. Please acknowledge the paper's limitations and develop practical implications loosely based on your literature review and empirical evidence.

I hope these comments help the authors improve their paper and encourage them to progress.

Reviewer 3 Report

Comments and Suggestions for Authors

Strengths of the Paper

1.       The paper introduces an approach, PDEC, to enhance knowledge graph reasoning by addressing predicate polysemy, which is a crucial but often overlooked aspect in the field of knowledge graphs.

2.       Rigorous experimental evaluations are conducted to demonstrate the effectiveness and adaptability of the methodology, showing significant improvements in knowledge graph reasoning accuracy.

3.       The PDEC framework is presented as a versatile and generalized framework applicable to any reasoning model, offering a scalable and flexible solution to enhance performance across various domains and applications.

4.       The paper delves into the impact of predicate settings in knowledge graphs and emphasizes the importance of predicate decomposition in enhancing reasoning performance.

Weaknesses of the Paper

1.       While the paper discusses the performance improvement of PDEC over traditional baseline methods on different datasets, it could benefit from a more detailed comparison with existing state-of-the-art approaches in the field of knowledge graph reasoning.

2.       The paper could provide more detailed insights into implementing the PDEC framework, including the algorithms or techniques used for predicate decomposition and reasoning enhancement.

3.       Some technical aspects, such as the specific datasets used, experimental setup, hyperparameters, and evaluation metrics, could be further elaborated to enhance the reproducibility and transparency of the experimental results.

4.       The paper could address the scalability of the PDEC framework, especially when dealing with very large knowledge graphs or datasets.

Comments on the Quality of English Language

Minor editing to the English language is required.

Round 2

Reviewer 1 Report

Comments and Suggestions for Authors

The authors made the requested changes, and the article could now be a relevant contribution to the state of the art. For that reason, I think it is ready to be published.

Author Response

Dear reviewer,

Thank you very much for your guidance and support in our work. Based on the feedback of another reviewer, we have made minor revisions to the article, mainly regarding the introduction and related work sections.

Best regards,

Xin Tian

Reviewer 2 Report

Comments and Suggestions for Authors

This is the second review of the paper entitled "PDEC: A Framework for Improving Knowledge Graph Reasoning Performance through Predicate Decomposition," I have no choice but to acknowledge that the authors have made several improvements. Firstly, they set out the objectives more specifically. Secondly, they move the paper contribution to the conclusion section. Thirdly, they advance the paper structure description at the end of the introduction. Fourthly, they add limitations in the conclusion section.

However, there remain several shortcomings, as follows:

(1)  The authors muddle through the paper structure. The introduction is much more extensive than the literature review section, and the methodology is too extensive. I would like to know if the literature review section is enlarged up to the point of being double the introduction. Similarly, the methodology should be smaller and shorter than the analysis of the results section. That said, it might be not applicable in this field since paper structures vary depending on the discipline.

(2)  Barely does the literature section review the literature. Not only is it shorter than the introduction, but it is also devoid of hypotheses or theoretical propositions. It is nothing more than three short subsections. I would like to know if the authors make better use of these three subsections to highlight three theoretical propositions.

(3)  It is hard to see the internal connection between the literature review and the results section's analysis. The authors should keep the same structure between the literature review and the result sections to build three empirical contrasts in the result section rooted in the literature review.

(4)  The discussion does not compare the results obtained and those of other papers.

I hope these comments help improve the paper and encourage the authors to proceed.

Author Response

Dear review,

Reviewer 3 Report

Comments and Suggestions for Authors

The authors have satisfactorily replied to the concerns raised and complemented the manuscript accordingly.

Author Response

(The authors gave the same response as above.)
